# Ophthalmic Manifestations of the Monkeypox Virus: A Systematic Review and Meta-Analysis

**DOI:** 10.3390/pathogens12030452

**Published:** 2023-03-14

**Authors:** Aravind P. Gandhi, Parul Chawla Gupta, Bijaya K. Padhi, Mokanpally Sandeep, Tarun Kumar Suvvari, Muhammad Aaqib Shamim, Prakasini Satapathy, Ranjit Sah, Darwin A. León-Figueroa, Alfonso J. Rodriguez-Morales, Joshuan J. Barboza, Arkadiusz Dziedzic

**Affiliations:** 1Department of Community Medicine, ESIC Medical College & Hospital, Sanathnagar, Hyderabad 500038, India; 2Department of Ophthalmology, Postgraduate Institute of Medical Education and Research, Chandigarh 160012, India; 3Department of Community Medicine and School of Public Health, Postgraduate Institute of Medical Education and Research, Chandigarh 160012, India; 4School of Medical Sciences, University of Hyderabad, Telangana 500046, India; 5Medicine and Surgery, Rangaraya Medical College, Kakinada 533003, Andhra Pradesh, India; 6Department of Pharmacology, All India Institute of Medical Sciences, Jodhpur 342005, India; 7Department of Virology, Postgraduate Institute of Medical Education and Research, Chandigarh 160012, India; 8Department of Microbiology, Institute of Medicine, Tribhuvan University Teaching Hospital, Kathmandu 46000, Nepal; 9Department of Microbiology, Dr. D.Y. Patil Medical College, Hospital and Research Centre, Dr. D.Y. Patil Vidyapeeth, Pune 411018, Maharashtra, India; 10Datta Meghe Institute of Higher Education and Research, Jawaharlal Nehru Medical College, Wardha 442001, India; 11Facultad de Medicina Humana, Universidad de San Martín de Porres, Chiclayo 15011, Peru; 12Centro de Investigación en Atención Primaria en Salud, Universidad Peruana Cayetano Heredia, Lima 15102, Peru; 13Grupo de Investigación Biomedicina, Faculty of Medicine, Fundacion Universitaria Autonoma de las Américas, Sede Pereira, Risaralda, Pereira 660003, Colombia; 14Escuela de Medicina, Universidad Cesar Vallejo, Trujillo 13007, Peru; 15Department of Conservative Dentistry with Endodontics, Medical University of Silesia, 40-055 Katowice, Poland

**Keywords:** mpox, ocular manifestations, outbreaks, meta-analysis

## Abstract

Background: The accurate estimation of the prevalence of mpox-induced ophthalmic lesions will enable health departments to allocate resources more effectively during the ongoing mpox pandemic. The aim of this meta-analysis was to estimate the global prevalence of ophthalmic manifestations in mpox patients. Methods: A systematic search was carried out in seven databases—Pub Med, Scopus, Web of Science, EMBASE, ProQuest, EBSCOHost, and Cochrane—for studies published on or before 12 December 2022. The pooled prevalence of ophthalmic manifestations was estimated by the random effects model. Risk of bias assessment of the studies and sub-group analysis to explain heterogeneity were undertaken. Results: Overall, 12 studies were included, with 3239 confirmed mpox cases, among which 755 patients reported ophthalmic manifestations. The pooled prevalence of ophthalmic manifestations was 9% (95% confidence interval (CI), 3–24). Studies from Europe reported a very low prevalence of ocular manifestations of 0.98% (95% CI 0.14–2.31), compared to studies from Africa with a substantially higher prevalence of 27.22% (95% CI 13.69–43.26). Conclusions: A wide variation in the prevalence of ocular manifestations among mpox patients was observed globally. Healthcare workers involved in mpox-endemic African countries should be aware of ocular manifestations for early detection and management.

## 1. Introduction

The ongoing multi-country outbreak of mpox (formerly known as monkeypox) has been declared a “Public Health Emergency of International Concern” (PHEIC) by the World Health Organization (WHO). Monkeypox virus (MPXV) causes mpox, a zoonosis. The origin of mpox traces back to 1958, when MPXV was identified in monkeys at a laboratory in Denmark. Historically, in 1970, the first mpox patient was identified from the Democratic Republic of the Congo (DRC) [1]. From 1970 to 1990, there were sporadic disease outbreaks in Central and West African countries. Post 1990, mpox cases were on the rise, and the majority of the cases were from the DRC. Presently, mpox is endemic in countries of Central and West Africa [2]. The first mpox outbreak outside Africa occurred in the United States of America (USA). The current multi-country outbreak was first noted in the United Kingdom, followed by Portugal, Canada, and Spain. Following this, many other countries reported mpox outbreaks [2], most of which were not epidemiologically related to the mpox-endemic countries. As of 11 December 2022, the WHO reported 82,624 confirmed mpox cases spanning 110 countries. Although the overall risk assessment at the global level is moderate, the Americas are at high risk [3].

Mpox is largely a self-limiting disease, with a maximum incubation period of 21 days. However, mpox has been responsible for considerable fatalities. Historically, the mortality rate has varied from 1% to 10% in endemic countries [4,5], while it has been between 3% and 6% in the ongoing multi-country break. The genetic clade of MPXV also impacts the transmission and severity of the mpox disease. The Congo basin clade has been shown to have higher transmissibility and cause more severe disease than the West African clade [6]. Rash (93%) and fever (72%) are the most common clinical features reported among mpox patients [7]. The clinical manifestations of mpox are similar to smallpox [2], with lymphadenopathy being the differentiating feature in mpox [8]. The epidemiological profile of mpox is evolving, showing variations in risk profile, clinical characteristics, and disease outcomes [9,10]. This transition is under study during the ongoing multi-country outbreak [7]. Atypical manifestations involving other systems, such as the oral, neurological, and respiratory systems, have been reported among patients with mpox [7,11]. Similarly, ocular signs and symptoms have also been reported among mpox patients. The manifestations have ranged from mild rash and focal lesions in the peri-ocular areas to visual loss [12]. Conjunctivitis, peri-ocular vesicular rash, photophobia, and oedema have been reported as the most common ocular manifestations (>20%) [12]. Among these, conjunctivitis was reported among children less than ten years old [13]. Atypical presentations of mpox are not uncommon. An mpox case from Spain had lacrimation, eye pain, and photophobia as the initial and presenting symptoms [14]. Conjunctivitis has been associated with an increased rate of bed riddance among mpox patients [13]. Conjunctivitis has also been hypothesised as a potential route of human-to-human transmission of mpox [14]. As sequelae, corneal ulcerations (1–4%), keratitis (3.6–7.5%), and, finally, vision loss (5–10%) have been recorded in previous studies [15,16]. Thus, the morbidity associated with ophthalmic lesions in patients with mpox is substantial. 

Ocular manifestations and complications are more prevalent among unvaccinated (mpox/small pox vaccine) individuals [12,15]. Considering the current scenario where vaccine protection against mpox in older adults is also waning due to time lapse, ocular manifestations may lead to severe complications. Therefore, it becomes essential to quantify the prevalence of ophthalmic manifestations in mpox. This will inform healthcare planners in anticipating and mobilising resources to suspect, identify, and manage ocular manifestations at the earliest opportunity and prevent complications. Based on our current evidence, no meta-analysis of ocular manifestations among mpox patients could be found while searching electronic databases. Hence, the following systematic review with meta-analysis was conducted to explore the ocular features of mpox and estimate the pooled prevalence of mpox-associated ocular manifestations, globally and regionally.

## 2. Materials and Methods

### 2.1. Research Question and Selection Criteria

The present systematic review and meta-analysis were carried out based on the following research question: “What is the prevalence of ocular lesions in mpox patients?”. The “preferred reporting standard of systematic reviews and meta-analysis” (PRISMA) checklist was adhered to in the index meta-analysis (Appendix A). The systematic search and identification of eligible studies were centred on the PICO criteria elaborated in (Appendix A. The current systematic review and meta-analysis protocol has been registered with the International Prospective Register of Systematic Reviews (PROSPERO), with reference ID CRD42022383265. 

### 2.2. Databases Included and Search Strategy 

The search was carried out in the following seven databases on 12 December 2022: PubMed, Scopus, Web of Science, EMBASE, ProQuest, EBSCOHost, and Cochrane (Appendix A). We also searched pre-print servers, such as medRxiv, arXiv, bioRxiv, BioRN, ChiRxiv, ChiRN, and SSRN. Furthermore, studies obtained by hand search in the references of eligible primary research papers and reviews, which met our eligibility criteria, were also included in the data extraction. The search keywords included were “mpox”, “MPXV”, “Monkeypox”, “ophthalmic”, “eye”, and “ocular”. The database-wise search strategy was applied, and the results obtained are enumerated in Appendix A. The identified articles were managed using Mendeley Desktop V1.19.5 software. 

### 2.3. Selection of Studies 

#### 2.3.1. Title Abstract Screening

The title abstracts of the articles found via the afore-mentioned systematic search were individually examined by two investigators (TKS and SM) by applying the eligibility criteria, and articles were identified for full-text screening. The co-investigators discussed the issue and came to a decision if there was a dispute about whether to include an article for full-text review.

#### 2.3.2. Full-Text Screening and Data Extraction

Eligible full-text articles were reviewed for suitability of data extraction by two investigators, and extraction of the data was performed independently (AGP and SM). In a consensus conference conducted after the independent extraction, discrepancies in the data extraction between the investigators were eliminated. The third investigator (BKP) adjudicated any discrepancies that could not be resolved. A final table was formulated that included information such as the name of the first author, publication year, the geography of the study where it was undertaken, design of the study, total mpox positive patients, and patients with ocular manifestations. PRISMA flow chart was used to report the general search, screening, data extraction, systematic review, and meta-analysis conducted to ensure scientific precision (Figure 1).

#### 2.3.3. Quality Assessment

Two investigators (AGP and SM) used the “National Heart, Lung and Blood Institute” (NHLBI) quality assessment method for case series and cross-sectional studies to independently evaluate the risk of bias in the included studies. 

### 2.4. Data Analysis

The pooled estimate of the ocular lesion prevalence, along with the 95% confidence interval (CI), was estimated by collating the total number of mpox patients and those with ocular manifestations. A sensitivity analysis was planned to account for the risk of bias in the studies by including only the studies rated as fair or good quality. Another sensitivity analysis was conducted to account for the potential overlap of the cases between various studies reporting from the same place and time. The pooled estimate was calculated after removing the potential studies with overlapping data if their country/region was the same and there was even a slight overlap in their period of data collection. To determine the cause of heterogeneity, we undertook the following subgroup analyses: geographical factors (according to the continent of the study), MPXV endemicity (endemic vs. non-endemic nations), and 2022 studies vs. pre-2022 studies. Heterogeneity between the studies was assessed by Q-statistics and I^2^ test. If the included studies were homogenous, then the Q value would be same as that of the degree of freedom (df). Depending on the I^2^ value, heterogeneity can be declared low (25%), moderate (25–50%), or high (>50%). Since there was high heterogeneity among the studies, a random effects regression model (DerSimonian and Laird estimator) was applied to determine the pooled estimate [17]. Prediction interval was calculated based on the Tau^2^ statistics [18]. Baujat plot, influence diagnostics, and leave-one-out analysis were applied to identify and address the outliers among the studies. A Doi plot was used to evaluate the publication bias. A trim-and-fill test was undertaken if there was a publication bias. Small study effects were assessed by the Eggers test. A *p*-value of <0.05 was interpreted as statistically significant. Comprehensive meta-analysis v4 software was used to conduct all the statistical analyses [19].

## 3. Results

### 3.1. Eligible Studies 

Figure 1 shows the selection process of the article as a PRISMA flow chart. The systematic search yielded 236 articles after removing 61 duplicates. After the title and abstract screening, 83 articles were included for full-text review. In the full-text review, 73 articles were found to be ineligible due to incorrect outcomes (35), incorrect study design (27), and incorrect patient population (11). Finally, 12 studies were found to be eligible for data extraction and meta-analysis. 

### 3.2. Study Characteristics 

The studies that are included were carried out between 1987 and 2022. Among the twelve studies included, six were cross-sectional studies [13,16,20,21,22,23], two were case series [24,25], two were retrospective studies [15,26], one was a prospective observational study [27], and one study was mentioned as a prospective cross-sectional study [28]. The studies had sample sizes between 21 [23] and 1057 [20]. Most of the included studies were conducted in the Democratic Republic of the Congo (six out of twelve studies (50%) (Table 1). 

The highest prevalence of ocular manifestations (51%) was reported by a study from the DRC [20], while the lowest prevalence was reported by a multi-continent study (0.57%) [24].

The heterogeneity among the studies was assessed to be high (I^2^ = 97%; *p* < 0.001) (Figure 2a). Hence, the random effects model was applied to determine the pooled prevalence. 

### 3.3. Pooled Prevalence

The meta-analysis included 3239 confirmed mpox cases, among which 755 patients reported ophthalmic manifestations. The pooled prevalence of ophthalmic lesions in the mpox patients was 9% (95% CI, 3–24) (Figure 2a). In the identification of potential outliers, the tests were able to identify two studies with a large impact on the cluster make-up: study 11 and study 12. (Appendix A). The sensitivity analysis after removing the outliers yielded a prevalence of 10% (95% CI 4–24), I^2^ = 88% (Figure 2b) (Table 2).

### 3.4. Risk of Bias

Nine studies were rated as fair or good quality based on the risk of bias assessment (Appendix A. The sensitivity analysis conducted with studies of good or fair quality studies (nine) yielded a pooled prevalence of 8% (95% CI 2.0–26) (Figure 2c) (Table 2), which was close to the overall prevalence (9%). 

The sensitivity analysis conducted after removing the studies with potential overlap of the cases revealed a pooled prevalence of 16% (95% CI 4–46) (Figure 2d) (Table 2). 

The Doi plot (Figure 3a) shows a symmetrical distribution of the studies included in the meta-analysis, which was further corroborated by using the Egger’s statistics (0.18, *p* = 0.874). Figure 3b shows the meta-regression in the form of a bubble plot, which indicates that the prevalence of the ocular manifestations was directly proportional to the sample size of the included studies.

### 3.5. Subgroup Analysis

Based on the geography where the study was conducted, studies from Europe reported a very low prevalence of ocular manifestations of 0.98% (95% CI 0.14–2.31), while studies from Africa reported a higher prevalence of 27.22% (95% CI 12.69–43.26). Similarly, the prevalence of ocular manifestations differed significantly according to the endemicity of mpox, with the mpox-endemic countries having a higher prevalence (27.22% (95% CI 13.69–43.26) than the non-endemic countries (1.05% (95% CI 0.09–2.68)). The prevalence of ocular lesions has been lower during the ongoing multi-country outbreak (0.61% (95% CI 0.13–1.31)) than in the cases reported before the 2022 outbreak (24.78% (95% CI 12.41–39.62)). The subgroup analysis based on the period of occurrence eliminated the heterogeneity among the studies which reported cases from the ongoing 2022 outbreak (I^2^ = 0%; *p*-0.55). Subgrouping by endemicity reduced the heterogeneity among studies from non-endemic countries (I^2^ = 53.93%; *p*-0.07), though it remained marginally on the higher side. However, subgrouping based on geography (continent) did not reduce the heterogeneity (Table 3).

The site-wise prevalence of ophthalmic manifestations and complications is enumerated in Table 4. 

The most affected site in the eye among mpox cases was the conjunctiva, with 11 of 12 studies reporting conjunctival lesions. Among them, data were available from six studies, which revealed a pooled prevalence of conjunctivitis of 13.89% (95% CI 6.92–22.67). Eye rash and conjunctival lesion (unspecified) were found to have a pooled prevalence of 14.37% (95% CI 6.91–23.71) and 1.62% (95% CI 0.80–2.69), respectively. Pustules or pseudo-pustules on the eyelids (1.08%), conjunctival ulcers (2.13%), redness, pain, and discharge in the eye (9.26%) were the other manifestations. The most common ophthalmic complication reported was photophobia in two studies [15,20], with the highest pooled prevalence of 30.87% (95% CI 28.13–33.67). Keratitis [21,22], corneal ulceration [16,21,22] (3.33%), corneal opacities (38.1%) [23], vision loss [16] (7.69%), and vision changes (2.31%) were the other complications reported in the included studies (Table 4). None of the patients from non-African countries reported vision loss. 

## 4. Discussion

In mpox patients, the pooled prevalence of ophthalmic manifestations has been determined at 9%, globally. Previous analyses based on a smaller number of studies found that the prevalence of ocular lesions ranges from 0.09 to 5% [7,29]. Although this is lower compared to manifestations involving other systems, such as rash, fever, and lymphadenopathy, one in ten patients with mpox could have ophthalmic lesions. Studies conducted among African mpox patients had a significantly higher prevalence of ophthalmic lesions than their European counterparts. A similar geographical variation in the prevalence of rash (the most common symptom of mpox) and oral lesions was also noted from previous reviews [7,11]. This is also closely related to the endemicity of mpox, as all African studies included were from mpox-endemic countries, and all studies outside Africa were from non-endemic countries. In terms of the time period, the ongoing outbreak of mpox showed a relatively lower prevalence of ophthalmic lesions than previous studies. Thus, there is a significant difference in the prevalence of ophthalmic lesions according to geography, endemicity, and the time period of cases.

The conjunctival lesion was the most common manifestation, photophobia was the most common complication, and vision loss was the most severe complication. The presence of conjunctivitis has been shown to have an association with other symptoms of mpox, including systemic and oral lesions [13]. Conjunctivitis has been reported as a factor of hospitalization [28] and a predictor of bed riddance among children [13]. Thus, ocular lesions may have a prognostic value in the outcomes of mpox. Corneal ulcerations, in combination with secondary bacterial infections, have been associated with severe ocular complications, including blindness [29]. Topical trifluridine, when administered together with tecovirimat for mpox patients with varying ophthalmic lesions, has improved the condition without complications [30]. Vision loss as a complication was reported only among endemic African countries, which could be due to the lack of adequate and appropriate treatment. Less severe ocular complications in non-African countries might also be due to the isolation of the less severe variant of MPXV (West African clade) as a causal agent in these countries [31]. None of the studies included in our analysis discussed the impact of ocular lesions on outcomes such as duration of hospital stay and mortality. 

The source of the acquisition of mpox had a differential impact on the occurrence of ocular manifestations, with primary cases (contracted from animals) having an ocular lesion prevalence of 10%, while the secondary cases (contracted from humans) had a lower prevalence of 5%, including complications [16]. Isolation of MPXV in the conjunctiva of prairie dogs’ conjunctiva has been reported [32]. Considering the above reports, there is a high suspicion of animal-to-human transmission of mpox through the ocular route, which might cause a higher prevalence of ocular lesions and complications among such cases. The isolation of MPXV from the ocular vesicles and conjunctiva of mpox patients has been reported [14,33]. However, MPXV transmission through human ocular secretions was not reported in any of the reviewed studies. 

A study from the DRC reported a differential prevalence of ocular manifestations between vaccinated (13%) and unvaccinated (17%) people against smallpox [21]. Smallpox vaccination offers protection against mpox, which may have reduced the incidence of ocular manifestations in the vaccinated.

The pooled estimate of the ocular manifestations of mpox patients has been assessed systematically for the first time in the index meta-analysis. Risk of bias assessment has been performed for the included studies using standard tools, and the robustness of the result was improved by conducting a sensitivity analysis after excluding the poor-quality studies. The heterogeneity of the included studies in the pooled prevalence was high, which is a limitation. We addressed the heterogeneity by conducting subgroup analysis accordingly, identifying geography and time of occurrence of cases as the potential factors for heterogeneity. Nevertheless, heterogeneity was high among African studies regardless of subgroup assessment. This could be due to the variations in sex, the genetic strain of the virus (clade), age groups of the patients reported in those studies, and/or other unidentified confounders. Therefore, pooled analysis findings must be interpreted with caution. Additionally, different diagnostic techniques employed in the different countries could also affect the proportion of ophthalmic manifestations in the studies included. Although we attempted to factor in the overlap of the cases among the studies by conducting a sensitivity analysis, an accurate estimate of the ocular manifestations might be elusive in the current analysis. 

In conclusion, the pooled prevalence of ocular manifestations among the mpox patients is 9%. There is a high variation in the prevalence of ocular manifestations according to the geography of the patients, with a higher prevalence of ocular manifestations found in endemic African countries. Healthcare workers involved in managing mpox in these African countries must be educated on the ocular manifestations of mpox for early detection and management. This, in turn, may prevent ocular complications, including vision loss. Robust strategies to address the overlap of cases from multiple studies need to be identified and applied in future reviews. Studies should report on the modality utilized for the diagnosis of the ophthalmic morbidity, which in turn will improve the comparability and pooling of the results from the studies. Furthermore, studies on the conjunctiva being a potential route of mpox transmission, therapeutics, and the prognostic role of ocular manifestations on the outcomes of mpox in endemic and non-endemic countries must be undertaken.

## Figures and Tables

**Figure 1 pathogens-12-00452-f001:**
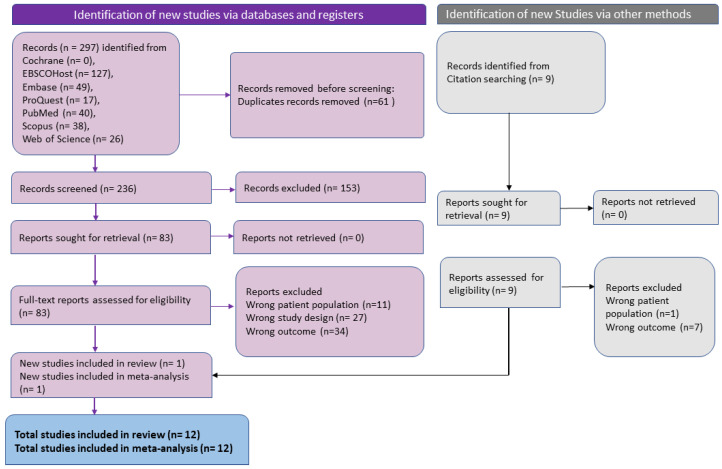
PRISMA flowchart for included studies in systematic review and meta-analysis of ophthalmic manifestations of mpox.

**Figure 2 pathogens-12-00452-f002:**
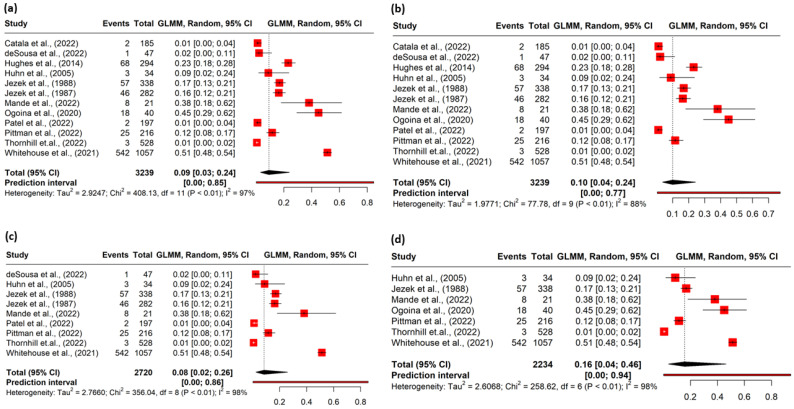
(**a**) Forest plot of the pooled magnitude of ophthalmic manifestations among mpox cases. (**b**) Forest plot of sensitivity analysis conducted after excluding outliers. (**c**) Forest plot of sensitivity analysis conducted after excluding poor quality studies. (**d**) Forest plot of sensitivity analysis conducted after excluding studies with potential overlap of cases.

**Figure 3 pathogens-12-00452-f003:**
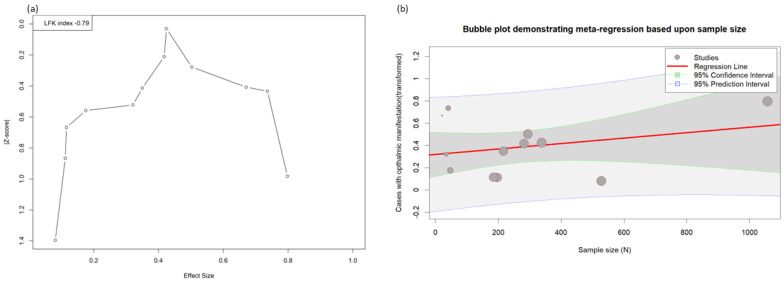
(**a**) Doi plot of the ophthalmic manifestations in the mpox. (**b**) Meta-regression showing the effect size with sample size.

**Table 1 pathogens-12-00452-t001:** Baseline characteristics of mpox patients presenting with ophthalmological manifestations (N = 12 studies).

Authors (Year)	Study Design	Geography	Sample Size	Prevalence (Ophthalmic Lesion)	Cluster	Key Findings
Catala et al., (2022)	Cross-sectional	Spain	185	1%	Patients from the national surveillance database system.	Pustules or pseudo pustules were noted on eyelids in two patients. Complications: pain, dysphagia, and conjunctivitis were reasons for hospitalization.
De Sousa et al., (2022)	Retrospective observational	Portugal	47	2%	Individuals with confirmed mpox infection.	Only one patient had palpebral conjunctiva ulceration.
Hughes et al., (2014)	Cross-sectional	Democratic Republic of Congo	294	23.1%	Real time PCR mpox positive patients.	A total of 23.1% of the mpox patients had conjunctivitis, and 47% of the conjunctivitis patients were bed-ridden.
Huhn et al., (2005)	Cross-sectional	United States	34	9%	Patients with confirmed mpox in medical records.	Around 9% of cases had involvement of conjunctiva (eye), and only one case had involvement of cornea, i.e., presented with keratitis and corneal ulceration.
Jezek et al., (1987)	Cross-sectional	Democratic Republic of Congo	282	13.4%	Mpox-diagnosed patients.	Forty-six were found to have opthalmic lesions (focal lesions). Keratitis, corneal ulceration was reported among 12 patients.
Jezek et al., (1988)	Cross-sectional	Democratic Republic of Congo	338	25%	Mpox-diagnosed patients.	Fifty-seven patients were reported to have conjunctivitis; eleven had corneal ulcers. Deforming scars, weak vision, and unilateral and bilateral blindness were observed in primary cases (10%) and secondary cases (5%) (29 cases).
Mande et al., (2022)	Cross-sectional	Democratic Republic of Congo	21	38%	Mpox-positive patients.	Of the 21 positive patients, 8 reported ocular lesions/corneal opacities.
Ogoina et al., (2020)	Retrospective study	Nigeria	40	45%	Hospitalised mpox-infected patients.	Eyes rashes were seen in 25% of 35 of the patients who gave details of their first symptom (9 patients). Nine patients reported conjunctivitis and photophobia.
Patel et al., (2022)	Descriptive case series	United Kingdom	197	1%	Patients confirmed with MPXV with a polymerase chain reaction.	Out of 197 patients, only 2 had conjunctivitis.
Pittman et al., (2022)	Prospective observational	Democratic Republic of Congo	216	18%	MPXV-specific PCR-positive patients.	Twenty patients had eye manifestations, which included eye redness, eye pain, and eye discharge. Five patients (2.3%) had reported visual changes. Conjunctival and other lesions were experienced by 14 patients.
Thornhill et.al., (2022)	Case series	America, Europe, Israel, Australia	528	1%	Confirmed human mpox infection cases from 16 countries.	Three patients had conjunctival mucosa lesions.
Whitehouse et al., (2021)	Cross-sectional	Democratic Republic of Congo	1057	51%	PCR-confirmed mpox patients.	A total of 210 (20.7%) had conjunctivitis, and 332 (33.2%) had photophobia.

**Table 2 pathogens-12-00452-t002:** Pooled prevalence of the ophthalmic manifestations among mpox patients.

Analysis	Pooled Estimate	95% CI	*p* Value	95% PI	I^2^
Main analysis	0.09	0.03–0.24	<0.01	0–0.85	97%
Outliers removed *	0.10	0.04–0.24	<0.01	0–0.77	88%
Poor-quality studies removed **	0.08	0.02–0.26	<0.01	0–0.88	98%
Potential overlapping studies removed ^#^	0.16	0.04–0.46	<0.01	0–0.94	98%

* Thornhill et al. and Whitehouse et al.; ** Catala et al., Hughes et al., and Ogonia et al.; ^#^ Catala et al., de Souza et al., Hughes et al., Jazek et al. (1987), and Patel et al.

**Table 3 pathogens-12-00452-t003:** Sub-group analysis of the ophthalmic manifestations of the mpox patients.

Factors	Estimate (95% CI)	*p*	I^2^	P Subgroup
Endemicity				<0.001
Non-endemic	1.05 [0.09–2.68]	0.07	53.93%	
Endemic	27.22 [13.69–43.26]	<0.001	98.09%	
Waves of outbreak				<0.001
2022 multi-country outbreak	0.61 [0.13–1.31]	0.55	0%	
Pre-2022 outbreak	24.78 [12.41–39.62]	<0.001	97.84%	
Continent				<0.001
Europe	0.98 [0.14–2.31]	-	-	
Africa	27.22 [12.69–43.26]	<0.001	98.09	
North America	8.82 [3.05–22.96]	-	-	
Multi-continent	0.57 [0.19–1.66]	-	-	

**Table 4 pathogens-12-00452-t004:** Site-wise lesions and complications in eyes among the patients with mpox.

	Number of Studies	Pooled Prevalence ES	95% CI
Manifestations			
Conjunctivitis	6 *	13.89%	6.92–22.67
Conjunctival ulceration	1	2.13%	0.38–11.11
Conjunctival lesion (unspecified)	2	1.62%	0.80–2.69
Eye rash (location unspecified)	2	14.37%	6.91–23.71
Pustules or pseudo-pustules in eyelid	1	1.08%	0.30–3.86
Redness, pain, and discharge	1	9.26%	6.07–13.87
Complications			
Photophobia	2	30.87%	28.13–33.67
Keratitis/Corneal ulceration	3	3.33%	1.99–4.95
Corneal opacity	1	38.1%	20.75–59.12
blindness (Unilateral/Bilateral)	1	7.69%	5.30–11.03

* I^2^ = 94.62%, *p* < 0.001.

## Data Availability

Documents containing all extracted data have been made available in the manuscript and the accompanying Appendix A.

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
