# Peer review of "Ophthalmic Manifestations of the Monkeypox Virus: A Systematic Review and Meta-Analysis"

_pathogens, 2023, doi:10.3390/pathogens12030452_

Round 1
Reviewer 1 Report
1. Diagnostic methods in different countries also affect the reported Ophthalmic Manifestations should be noted in the limitations
2. In the sensitivity analysis, the author should evaluate the effect of eliminating articles with possible overlap. If there were overlapping publications from the same country in the same period were identified, only the one with the largest sample size was included. The author should address this issue in the method.
3. In the discussion, the author could possibly point out the deficiencies of the research and provide suggestion for the future research.
Author Response
Reviewer #1:
Comment 1: Diagnostic methods in different countries also affect the reported Ophthalmic Manifestations should be noted in the limitations.
Response 1: Thank you very much for pointing this out. We agree with the reviewer and have incorporated the point as limitation in the revised manuscript as follows:
“Also, different diagnostic techniques employed in the different countries could also affect the proportion of the ophthalmic manifestations from the studies included.” (Page 10, Lines 324, 325)
Comment 2: In the sensitivity analysis, the author should evaluate the effect of eliminating articles with possible overlap. If there were overlapping publications from the same country in the same period were identified, only the one with the largest sample size was included. The author should address this issue in the method.
Response 2: Thank you directing us towards this sensitivity analysis. We agree with the reviewer that there may be chances of overlaps between the cases from multiple studies. Based on the reviewer comment, when we reviewed the studies which were conducted in the same country, we found that no two studies had the exact same time period of data collection. However, we conducted a sensitivity analysis after the removing the potential studies with overlapping data, if their country/region was same and there was even a slight overlap in the period of data collection. As per the above criteria, among them, one study was conducted each in Spain, Portugal and United Kingdom. Another included study (Thornhill et al) was a multi-country study which included the above mentioned three countries and the period of data enrollment of the mpox patients was overlapping. Hence, we removed the three smaller studies and retained the Thornhill et al. Among the studies conducted in DRC, there is a potential overlap between Jazek et al (1987) and Jazek et al (1988), and there is a potential overlap between Hughes et al and Whitehouse et al. As per the reviewer’ suggestion we removed the smaller studies (Jazek et al (1987) & Hughes et al) during the sensitivity analysis. Thus, we conducted a sensitivity analysis with 7 studies, after removing potential overlapping smaller studies.
“Another sensitivity analysis was conducted to account for the potential overlap of the cases between various studies reporting from the same place and time. The pooled estimate was calculated after removing the potential studies with overlapping data, if their country/region was same and there was even a slight overlap in their period of data collection.” (Page 4, Lines 159-163)
“Sensitivity analysis conducted after removing the studies with potential overlap of the cases revealed a pooled prevalence of 16% (95% CI 4-46) (Figure 2c).” (Page 7, Lines 215,216)
Comment 3: In the discussion, the author could possibly point out the deficiencies of the research and provide suggestion for the future research.
Response 3: Thank you so much for the suggestion. We have included the deficiencies and recommendations at the primary study level as well as at the review level as follows:
“Also, different diagnostic techniques employed in the different countries could also affect the proportion of the ophthalmic manifestations from the studies included. Although we attempted to factor in the overlap of the cases between the studies by conducting a sensitivity analysis, accurate estimate of the ocular manifestations might be elusive in the current analysis.” (Page 10, Lines 324-328)
“Robust strategies to address the overlap of cases from multiple studies needs to be identified and applied in the future reviews. Studies should report on the modality utilized for the diagnosis of the ophthalmic morbidity, which in turn will improve the comparability and pooling of the results from the studies.” (Page 11, Lines 335-338)
Reviewer 2 Report
The authors performed meta-analysis using publications from seven databases to estimate the global prevalence of ocular manifestations in patients diagnosed with mpox disease. The study is well designed and attracts the attention of healthcare workers to the ophthalmic complications associated with mpox and promotes the early mpox detection and disease management as well.
Only minor corrections are needed:
Line 71: correct typing: Americans
Line 86-87: two version of a term was used (periocular vs peri-ocular), please use either one or the other along the manuscript
In Figure 1: it is not clear for me how was 12 study included in the analysis from the total publications that matched the criteria. Out of 83 article included for full-text review, 73 were found to be ineligible. How the additional publications was chosen for analysis?
In Table 1, please use mpox instead of monkeypox (Hughes et al. 2014)
Author Response
Reviewer #2:
Comment 1: The authors performed meta-analysis using publications from seven databases to estimate the global prevalence of ocular manifestations in patients diagnosed with mpox disease. The study is well designed and attracts the attention of healthcare workers to the ophthalmic complications associated with mpox and promotes the early mpox detection and disease management as well.
Response 1: Thank you so much for the appreciation.
Comment 2: Only minor corrections are needed: Line 71: correct typing: Americans
Response 2: Thank you for the comment. However, here we are meaning the Americas region (North and South America), according to the WHO regions.
Comment 3: Line 86-87: two version of a term was used (periocular vs peri-ocular), please use either one or the other along the manuscript
Response 3: Thank you for the comment. We have corrected it to peri-ocular. (Page 3, Line 90)
Comment 4: In Figure 1: it is not clear for me how was 12 study included in the analysis from the total publications that matched the criteria. Out of 83 article included for full-text review, 73 were found to be ineligible. How the additional publications was chosen for analysis?
Response 4: Thank you for the comment. We would like clarify that out of 83 articles eligible for full text review, 72 were removed for the following reasons: wrong outcome (34), wrong study design (27) and wrong patient population (11). Thus, 11 studies (83-72) were eligible from the databases, to be included in the review and analysis. Apart from the database search we also did search by other methods like citation searching and identified one new eligible study for inclusion into review and analysis. Thus, we got 12 studies for the systematic review and meta-analysis, as mentioned in Figure 1.
Comment 5: In Table 1, please use mpox instead of monkeypox (Hughes et al. 2014)
Response 5: Thank you for the suggestion. We have corrected it (Table 1)